

# A web scraping app for smart literature search of the keywords

Muhammed Ali Mutlu[1], Eyup Emre Ulku[2] and Kazim Yildiz[2]

[1] Head of Data Platforms, NTT DATA Business Solutions, Istanbul, Istanbul, Turkey
[2] Faculty of Technology, Computer Engineering Department, Marmara University, Istanbul, Turkey

## ABSTRACT

Detailed literature search and writing is very important for the success of long research projects, publications and theses. Search engines provide significant convenience in research processes. However, conducting a comprehensive and systematic research on the web requires a long working process. In order to make literature searches effective, simple and comprehensive, various libraries and development tools have been created and made available. By using these development tools, research processes that may take days can be reduced to hours or even minutes. Literature review is not only necessary for academic studies, but it is a process that should be used and performed in every field where new approaches are adopted. Literature review is a process that gives us important ideas about whether similar studies have been conducted before, which methods have been used before and what has not been addressed in previous studies. It is also of great importance in terms of preventing possible copyright problems in future studies. The main purpose of this study is to propose an application that will facilitate, speed up and increase the efficiency of literature searches. In existing systems, literature searches are performed by browsing search sites or various article sites one by one and using the search tools provided by these sites. It is simple to use, allows the entire World Wide Web environment to be searched, and provides the user with the search findings. In this study, we have implemented an application that allows the crawling of the entire World Wide Web environment, is very simple to use, and quickly presents the crawl findings to the user.

# INTRODUCTION

Literature review is a process that contributes significantly to the success of every innovative study, such as a project, article, or thesis. A literature review is defined as the process of researching a particular subject in detail among the available resources and systematically collecting data on that subject (*Snyder, 2019*). In other words, it determines the roadmap for the research subject. It is not a correct point of view to think of a literature review as only quoting and benefiting from sources. The literature review plays an important role in obtaining answers to important questions such as whether the subject of interest has been studied before, from which angles the subject is handled and by which

Corresponding author
Eyup Emre Ulku,
emre.ulku@marmara.edu.tr

methods it is carried out and, which problem or problems in the literature it offer a solution. In line with these answers, a basis for an innovative study can be formed (*Winchester & Salji, 2016*). The first step of the research process is to identify the problem and the keywords related to the it. Existing literature review methods are often time-consuming and may miss relevant studies. This study aims to develop an application that addresses these inefficiencies by automating the literature search process, thus enhancing both speed and comprehensiveness. The primary research question guiding this study is: How can web scraping and crawling techniques be utilized to enhance the efficiency and comprehensiveness of academic literature searches? In addition, it is very important to determine which areas of the researched subject are focused on and which areas are missing in previous studies (*Anastasiadis, Rajan & Winchester, 2015*). The materials and instruments we will utilize to do the literature study must be decided upon in the second phase. Identifying access to resources is also needed. Thus, the target resources and tools needed for the literature review are determined. The quality of the studies conducted is directly proportional to the reliability of the sources examined in the literature review. It is crucial to use reliable websites and databases to obtain the examined sources because of this. Reliable sources that are widely known should be chosen, and the author and publication environment of the source should be examined. All studies in all sources connected to the research issue should be substituted with scientific sources. You can use easily accessible online books, articles, theses, and encyclopedias that are written on the subject for this purpose. The resources to be researched can be determined by looking at the academic environments in which the resources are published and the number of reviews. After the problem and the reliable tools and resources to be used to examine this problem are determined, the literature review should be planned, divided into subsections and classified. The problem-solving methodology, the resources consulted, the comparison of the generated solutions and the identification of their commonalities are all covered in detail in each subsection. As a result, it is guaranteed that the investigation will be conducted in a methodical manner. A conclusion summarizing the circumstances, shortcomings, and findings of the literature should be included at the end of the literature review. Critical assessments of contemporary problems and approaches should be conducted by incorporating data from the sources consulted for the literature review. The deficiencies identified as a result of the literature review and the determination of how to eliminate these deficiencies are very important in supporting the innovative aspect of the researched subject. When the described literature review stages are followed sequentially and systematically, it will be determined exactly which gap in the literature will be filled by the study. The literature research should be conducted again till the study's conclusion in order to verify its veracity and creative elements. A thorough and accurate examination of the literature is crucial to the effectiveness of the studies that are conducted. How web scraping and crawling techniques can be used to improve the efficiency and comprehensiveness of academic literature searches is the primary research question guiding this study. This study presents a smart literature review application that intends to guarantee thorough, quick, and efficient review processes as well as the inclusion of reputable sources in the review. Search engines have made it easier and faster than ever to

access the materials you're looking for. Search engines are provided to users by virtual libraries, which have supplanted physical libraries. These World Wide Web search engines also provide access to the resources required for a literature study. While the user may quickly and readily access information from search engines, navigating the websites that provide results and finding the content on each one is a highly challenging task. Search engine literature reviews take a lot of time to complete. While browsing the sites, there can be crucial information in these manual processes that is missed but will improve the task. Numerous investigations are conducted with the aim of enhancing the methodical literature review procedure (*Waffenschmidt et al., 2019*; *Macura et al., 2019*; *Gusenbauer & Haddaway, 2020*; *Haddaway et al., 2018*). In recent years, studies aiming to automate literature review processes have been tried to be carried out so that researchers can perform these processes more quickly and efficiently (*Van Dinter, Tekinerdogan & Catal, 2021*; *Asmussen & Møller, 2019*). Studies using approaches such as text mining, natural language processing and machine learning are presented in order to develop easy-to-use and fast-running systems by automating the literature review processes (*Feng, Chiam & Lo, 2017*; *Zdravevski et al., 2019*; *Marshall & Wallace, 2019*). Various methods have been developed so that users can automate their research faster and more efficiently. The basis of these methods is web crawling and web scraping techniques. Web crawling and web scraping are the primary techniques that can be used as a basis for developing systems that will perform literature searches autonomously (*Haddaway, 2015*). Bots that are coded to scan the entire World Wide Web (WWW) environment to search the web and obtain the associated site addresses are defined as web crawler bots and bots that are used to extract meaningful information from the obtained site addresses are defined as web scraper bots. The whole process of combining these developed bots with appropriate algorithms and making them able to carry out operations systematically and automatically and extracting meaningful data is called web crawling and scraping. These search tools help to create structured databases by obtaining relevant data in a meaningful way (*Khalil & Fakir, 2017*; *Ferrara et al., 2014*; *Myllymaki, 2001*; *Uzun, 2020*; *Turk, Pastrana & Collier, 2020*). While these tools make data acquisition and processing easier, concerns have been raised about their legality and ethical implications (*Haque & Singh, 2015*). In particular, there have been drawbacks related to stealing various personal data or financial information from websites. A study was conducted by the Ethical Decision-Making and Internet Research Committee to use web scraping methods as a guide for use (*Markham et al., 2012*). As it is clearly revealed in this study, the extraction of financial or personal information by web scraping is considered a legal violation and all the country's courts have stated that it is against it. Website owners can also specify the parts of the sites that they allow and prohibit to be visited by bots on their sites with the file at the "website/robots.txt" path. This is also considered in the same context. Apart from this, web scraping is a very useful and legal method when there is no personal information violation and it is done by considering the "robots.txt" file of the site. Conducting a comprehensive and efficient literature review ensures a solid foundation for the upcoming work, thereby enhancing the quality and presenting an up-to-date and innovative study (*Xiao & Watson, 2019*). Literature review is at the forefront of the stages that should be carried out not only in academic studies but

also in every innovative study. It is also very important to avoid problems such as copyright and license violations. In this article, we have used web crawling and web scraping techniques to develop a data extraction application from popular article websites, with due regard to the legal framework. We developed an application that quickly and automatically scans the keyword entered through a simple interface, taking into account the search criteria, in predetermined popular article websites in the WWW environment, and obtains the abstracts of the publications it finds. In this way, the user will be able to perform the literature review, which he/she will do before carrying out an academic study, in a much shorter time and in a much more comprehensive way by using this application. Thus, it will be possible to quickly access the abstracts and other keywords of the studies related to the study subject. As a result, by using these technologies, it is aimed to minimize the time to obtain information for literature review and to maximize efficiency. The developed application offers the opportunity to perform literature review processes in a shorter time, more comprehensive and more efficient way. We can summarize the contributions of the developed application to the literature review processes under four subheadings.

- **Time efficiency:** By automating the initial screening process, it significantly reduces the time researchers spend searching and selecting literature. Researchers can search for studies in multiple academic databases at the same time according to the criteria they define. As a result, users can quickly obtain summary data of studies found in the literature according to predefined criteria.
- **Comprehensive coverage:** It provides wider access to studies by overcoming the limitations of manual searches that can miss relevant research. Thus, it enables the researcher to have a more comprehensive command of the literature on the subject under study.
- **Facilitates the review process:** Allows researchers to quickly collect and review a larger set of abstracts, providing more time for in-depth analysis of selected articles.
- **Increases accessibility:** It makes it easier for researchers to discover and access a wide range of publications, including those outside the immediate search parameters.

This approach does not replace a thorough reading and analysis of selected articles, but rather enhances the efficiency and scope of the literature review phase.

To help readers navigate the article, we provide a brief overview of its structure. "Related Works" reviews existing literature on web scraping and web crawling applications, focusing on their use in various fields such as health, social media, finance, and marketing, and highlights the gap our study aims to fill in the context of literature review processes. "Material and Methods" details the development of our web scraping application, including the modular structure, the design of the web crawler and scraper, the backend and frontend development, and the legal and ethical considerations. In "Results and Discussions", we explain the usage of the application and exemplify a case in the output section. Finally, "Conclusion" summarizes the main contributions of our study, discusses the practical applications and benefits of our web scraping tool, and outlines potential future enhancements and areas for further research.

## Objectives

The primary aim of this study is to develop and evaluate a web application that leverages web scraping and crawling techniques to enhance the efficiency and comprehensiveness of academic literature searches. By addressing the limitations of existing literature review methods, this study seeks to provide a more effective and user-friendly tool for researchers. The specific objectives of this study are as follows:

- To develop a web application that automates the literature search process using web scraping and crawling techniques.
- To evaluate the efficiency of the proposed application by comparing the time taken to complete literature searches with traditional methods.
- To assess the comprehensiveness of the literature search results obtained using the application compared to traditional manual searches.
- To provide a user-friendly interface that allows researchers to easily and quickly access relevant literature.
- To increase accessibility to a broader range of academic publications by overcoming the limitations of manual search methods.
- To reduce the time and effort required for conducting comprehensive literature reviews, enabling researchers to focus more on analysis and synthesis.

## RELATED WORKS

The development of web crawling and web scraping applications has significantly advanced the field of data collection and processing. Various studies have employed these techniques to facilitate efficient data gathering for different purposes, ranging from semantic web solutions for health information to news aggregation and recommendation systems. This section reviews existing applications and highlights the unique features and advantages of our proposed application in comparison to these studies.

Nowadays, with the developments in technology, the diversity of electronic media and the use of these media are becoming more and more widespread (*Dwivedi et al., 2021*). The widespread use of electronic media causes a rapid increase in the amount of data stored and in circulation. Search engines offer the opportunity to access a large amount of data related to the searched topic (*Sheela & Jayakumar, 2019*). Ease of access to large amounts of data provides a significant advantage, but also complicates the process of obtaining meaningful data that will be useful to us (*Sivarajah et al., 2017*; *Patel, 2019b*, *2019a*). In this direction, in recent years, many studies have been carried out using many different methods in order to obtain meaningful data from large amount of data (*Patel, 2019a*; *Jin, Xing & Wang, 2020*; *Pandey & Shukla, 2018*; *Liang, 2020*). In an article researched at the University of Sumatera Utara, which highlighted the seriousness of tropical diseases, they found that people living in Indonesia rely on search engines to find treatments for tropical diseases, such as medicines and treatment methods. However, they explained that traditional search engines return results by taking into account the synonym of the disease when searching for treatments for diseases. They mentioned that to overcome this

problem, a search engine with semantically correct results should be developed as a solution. This solution, which they call the semantic web, reveals the relationship of the data to its synonyms using a form called Resource Description Framework (RDF). Initial research in Indonesia was unsuccessful because the Ministry of Health websites could not provide files in RDF format. Later, the study aimed to transform it into a platform that produces health information in RDF format using web scraping technologies. In this way, web scraping technologies were used to collect and extract words from popular health websites, to establish the relationship between ontology and terms, and as a result to produce information in the form of RDF (*Amalia, Afifa & Herriyance, 2018*). A platform called Newsone has been developed at Agni Technology College, India, which collects the latest news updates from national and international sources and summarises them concisely. The aim of this application is to provide quick results that save time and effort in finding the latest news. Web scraping and crawling techniques were used. To describe the method, application administrators store RSS addresses in the database. The web scraping bots in the application also dynamically and periodically crawl the relevant RSS sites and store the relevant content. In the next step, the extracted data is documented by category and URL, and a model is created. Thanks to this model, users can access the news according to their interests and relevance, or they can perform category-based searches. In other words, with this developed application, the reader can easily and quickly access the content of more than 100 licensed and reliable news sites worldwide (*Sundaramoorthy, Durga & Nagadarshini, 2017*). *Ertam (2018)* from Firat University in Turkey emphasised the importance of collecting and processing data from websites. In this study, categorical news headlines and summaries on a Turkish news agency website were collected using web scraping methods, and the test data was classified using the "one hot encoding" process, one of the vector classification approaches, with tensorflow-based deep learning methods. As a result of the classification, an accuracy rate of over 90% was achieved (*Ertam, 2018*).

In the article written by *Junjoewong, Sangnapachai & Sunetnanta (2018)* at Mahidol University, they mentioned that people spend a lot of time on websites to find promotions and campaigns that are relevant to them. The mobile application they developed, ProCircle, uses web scraping technology to collect promotions on a single platform. The mobile application they developed, ProCircle, used web scraping technology to collect promotional news on a single platform. They also crowdsourced the publicity news they collected. Thanks to this support, they enabled users to scan promotions en masse (*Junjoewong, Sangnapachai & Sunetnanta, 2018*). The article published at the 2020 IEEE European Symposium examined the defence mechanisms of websites against web scraping techniques. It was mentioned that some website administrators try to prevent data collection from their sites as a defence technique, while others make it difficult to collect data. The defence techniques used in the related article were examined, as well as the defence-breaking techniques developed against them. This was done in an attempt to determine the success of defence techniques. According to the results of the study, they have proven that even if these defence techniques slow down scraping, there is always a scraping solution in some way (*Turk, Pastrana & Collier, 2020*). Using web scraping technologies, a study was conducted in Indonesia on the product promotion and

marketing features of Instagram, one of today's popular social media platforms. It was pointed out that when trying to buy the product, it is necessary to have an account approved by Instagram to be trusted. There are too many accounts with fake followers. It was stressed that there is a need for a system that makes this determination, as users often cannot understand which account is fake. In this study, it was ensured that the parameters taken into account, such as the number of followers on the Instagram account, the number of likes, the number of comments and the number of recent posts, were sufficient to identify the account as a real marketer account using web scraping techniques. It was reported that it was successfully investigated with a rate of 75% (*Akrianto, Hartanto & Priadana, 2019*). A study in India highlighted the fact that news on the internet can be fake. They proposed a model to detect fake information and news using deep learning and natural language processing methods. The data set of news content obtained from news sites using web scraping methods was trained with deep neural networks and the correlation of words in the related documents was found using natural language processing. It was stated that these correlations serve as a starting value for the deep neural network and help to understand whether the news is fake. It is stated that the classification is done using recurrent neural network, long short-term memories and graded recurrent units methods. It is emphasised that a good training set is mandatory, as fake news can be detected according to the training model, and it is highlighted that the results of this study met this requirement and gave successful results (*Verma, Mittal & Dawn, 2019*). A study on health was conducted at the Karunya Institute of Technology in 2019. According to the study, it was highlighted that the World Health Organization reported that cancer is the second leading cause of death in the world. They stressed that people battling cancer often experience negative emotions such as anger, anxiety and depression, which have a negative impact on their disease. Karunya University, which carried out the study, has been working on the design of a chatbot (messaging bot) where these people can easily talk and ask all their questions and satisfy their need to communicate with people who have the same feelings. This chatbot, which can only be used by people with cancer, is designed to answer questions about cancer, such as treatment, symptoms and psychological distress. During the development of the chatbot, many cancer forums, which are a rich source of information on the subject, were trained using data obtained by web scraping methods. By using sentiment analysis methods, the aim is to determine the current emotional state of the user and to make the bot behave like a human and provide emotional relief to the user (*Belfin et al., 2019*). According to a Turkish study on global events that people can attend, more people would likely participate if these events were publicized on social media using an intelligent suggestion system. A system that can be implemented into numerous social media platforms and is used to provide recommendations to users has been developed in this study. Through the use of web scraping technologies, the system is able to acquire the online activities that a person has attended or plans to attend. The system functions by taking into account the user's location and social surroundings, making recommendations for suitable activities for them (*Kayaalp, Ozyer & Ozyer, 2009*). One study focused on how web scraping and natural language processing methods can produce solutions to complex problems in computer science education. It has been suggested that large amounts of data

from public web pages can be extracted by web scraping techniques and estimating more data similar to this data. In addition to this process, it has been studied how it can be used to extract salient information from this large amount of data using natural language processing methods. While putting these techniques into practice, an application has been put forward to examine the current trends in the job market for computer science students. According to a survey, grocery stores' ability to offer higher-quality, more affordable products with shorter lead times than their rivals boosts sales in today's cutthroat market. Japan's 54,000 grocery outlets were investigated for this. It has been suggested that the excessive number of supermarkets and their ongoing expansion could result in longer product supply and marketing periods. This study determines the best truck route for wholesalers who carry goods to many grocery stores by utilizing web scraping technologies and Excel Visual Basic. In order to examine the problem, firstly, the network flow model was drawn and the geographical data of all grocery stores, their warehouses and gas stations were taken with web scraping techniques. By integrating these geographical data with the Google API service, they developed a method to find the most optimum and fastest result (*Malik & Rizvi, 2011*). In the smart chef application developed by *Chaudhari et al. (2020)* the details of all recipes are extracted using the web scraping technique. Afterwards, an algorithm is presented that allows it to choose recipes that are suitable for the determined diet plan or based on ingredients (*Chaudhari et al., 2020*). In the smart application developed by *Dang et al. (2023)* provides a comprehensive examination of factors that predict the emergence of new outlinks during focused web crawling. The study employs a systematic approach to identify and analyze various predictors, with the goal of enhancing the efficiency and accuracy of focused web crawlers in discovering relevant and valuable new links. By examining the dynamics and characteristics of web navigation and link formation, the article contributes to the field of web mining and information retrieval, offering insights and practical strategies for developing more sophisticated and effective web crawling technologies (*Dang et al., 2023*). *Khalid et al. (2021)* present an innovative approach to improving scientific research methodologies. This study introduces a new system that combines inverted indexes and structured search techniques with citation network analysis to improve the relevance and accuracy of search results in academic databases (*Khalid et al., 2021*).

While several studies have leveraged web scraping and crawling techniques for various applications, our proposed work stands out due to its specific focus on academic research, incorporating advanced features such as citation analysis, keyword relevance filtering, and legality controls. These unique attributes enhance the efficiency, accuracy, and compliance of literature search processes, providing significant advantages. Our application thus represents a significant advancement in the field, offering researchers a powerful tool to streamline and enhance their literature review processes.

## MATERIALS AND METHODS

The application, developed in a modular structure, provides users with an easy-to-use interface. This interface allows users to search for keywords that will form the basis of their research, using various filters. A service has been created to ensure that the results of the

searches carried out are quickly presented to the users. The creation of this service was made possible by the development of web scraping and web crawling bots. Accordingly, the application was developed in four phases: web crawler development, web scraper development, backend development and frontend development.

To ensure the successful development and deployment of our web scraping application, we carefully selected a range of software tools tailored to meet the specific needs of our project. Each tool was chosen based on its unique capabilities and how well it complemented the various phases of our application development. Table 1 provides an overview of the software tools used in the study, detailing their purposes and the rationale behind their selection. This information is intended to enhance the reproducibility of our study by providing clear insights into our methodological choices.

## Browsing-web crawler

As a first step in obtaining the article data associated with a keyword entered, a crawler bot was developed to search the WWW environment and return results according to the desired criteria. The crawler bot was developed using the Google Search JSON API service, which is offered free of charge with certain restrictions by Google, the most popular search engine today. This service, offered by Google to users, is a request-based search service on the WWW using a dedicated and programmable search engine (*Google, 2023a*). After subscribing to the service, a user-specific API key is created. This API key is required for authentication on each request. After registration, a programmable search engine must be created. Certain configurations can be made in this dedicated search engine. Settings can be made such as searching only in the specified languages, in the specified regions, on the specified sites, or excluding these specified sites. This API has been customised for the application by making various changes. Exact match and pattern matching approaches were used together according to the criteria entered by the user *via* the interface. This dedicated search engine has an ID value that corresponds to the user's API key. Similarly, any request sent to the service must include this ID to indicate which search engine is being used. There are 10 URLs on a page for each request made with the word to be searched for on this search engine provided by Google, and it returns a maximum of 10 pages of results. In other words, 100 URLs are returned in total for a searched keyword, 10 pages and 10 URLs. In addition, the API key can be requested 100 times per day. If the limitations are not enough for the user, there is a fee of $5 per 1,000 requests (https://developers.google. com/custom-search/v1/overview; *Google, 2023b*). This fee is also limited to 10k requests. If needed, Custom Search Site Restricted JSON API offered by Google for a fee can be used for more than 10k requests (URL, d). Once the personalised search engine was created and a crawler bot was developed to work in our backend service, which was developed using the Python language. This crawler bot sends a request to the search engine and returns results according to the searched keyword, filters and the website to be searched. In order to perform a keyword-based search for the literature review, the search is limited to ResearchGate, IEEE, Springer, ACM sites, which are very popular for academic studies in engineering fields. The robots.txt files of these sites were examined in detail and it was confirmed that there was no illegal situation. Our web scraping application relies on

**Table 1 Software tools used in the study and their selection rationale.**

| Software tool | Purpose | Reason for selection |
|---|---|---|
| Python | Programming language for web scraping and data analysis. | Versatile, extensive libraries, readability and accessibility for developers of varying expertise |
| BeautifulSoup | Parsing HTML and XML documents for data extraction. | Powerful capabilities in parsing, essential for extracting data from web pages. |
| Requests | Handling HTTP requests to interact with web pages and retrieve content | Efficient handling of HTTP requests, well-documented |
| Google search JSON API | Retrieving relevant URLs based on user-defined keywords | Reliable, comprehensive search capabilities, easy integration with web scraping framework |
| FAST API | Building backend services | Speed, ease of use, automatic generation of interactive API documentation |
| React | Developing frontend interfacem | Flexibility, efficiency, strong community support |
| Redux | State management in conjunction with React | Effective state management, ensures seamless user experience |
| Cloud firestore | Database solution for managing and storing data | Scalability, flexibility, seamless integration with other tools |

publicly available data from specific academic databases. Some relevant studies might be behind paywalls or restricted access, which could limit the comprehensiveness of our data collection. In particular, sites such as Scopus, ScienceDirect, and Academia have robot blocking barriers or require paid access, which prevents us from including them in our data sources. The crawler bot will work with the search for the desired keyword, and as a result the associated results will be listed on the article sites using the Google Search JSON API.

## Exact match and pattern matching

To enhance the efficiency and comprehensiveness of the literature search process, we employed two distinct approaches: Exact Match and Pattern Matching. These methods were implemented using Python libraries to ensure precision and flexibility in retrieving relevant academic documents. For the Exact Match approach, we utilized built-in string operations, which are simple and efficient for exact keyword matching, and the re library (regular expressions) to provide additional control and ensure precise matches by treating the keyword as a regular expression pattern. The Pattern Matching approach was primarily implemented using the re library to compile and search with regular expressions, allowing for the identification of variations of the keyword, including synonyms and related terms.

## Data collection/web scraper

The articles' abstracts from the url list will be extracted using a web scraper bot that has been constructed as a second stage. Python has been selected since it offers many libraries on web scraping technologies and will be integrated into the backend service that will be constructed.

### Legality control

Site owners may not want their sites to be visited by crawling and scraping methods. And in this context, they have the right to take legal action against those who visit their sites for

this purpose. They indicate this request with the robots file they have on their site. On the sites there is a file about where and how scraping and crawling bots can navigate on the site *via* the "domain/robots.txt" path. For legal crawling and scraping work, it is necessary to take action on the site by paying attention to this file. An example of the robots.txt file taken from Researchgate.net is shown in Fig. 1. According to this, web paths written with the allow command can be visited by any scraper or browser bot, while paths written with the disallow command should not be visited. As mentioned above, the "robots.txt" file has been taken into account to ensure that scraping is legal. Before browsing the obtained pages, an algorithm has been developed to check that there is no problem in scraping the article on the page related to the "reppy" library using web scraping methods. According to the robots.txt algorithm, if there is an obstacle, the page is avoided. If there is no obstacle, a "successful" message is returned. This algorithm is always used before scraping algorithms (https://pypi.org/project/reppy/; *Python Software Foundation, 2023c*).

To ensure our web scraping activities are legal and respect the preferences of website owners, we developed a legality control algorithm. This algorithm checks the robots.txt file of each target website before starting any scraping operations. The robots.txt file specifies which parts of the website are allowed or disallowed for crawling by bots.

The legality control algorithm involves several key steps to ensure compliance with the robots.txt file of each website before initiating any scraping activities. These steps include importing the necessary library, extracting the base URL, fetching the robots.txt file, checking the permissions for crawling, and returning the result. Below is a detailed explanation of each step in the algorithm:

1) **Import the Library:** We use the **reppy.robots** library to handle the fetching and parsing of the robots.txt file. This library simplifies the process of checking permissions.

2) **Extract Base URL:** The function **extractBaseUrl** is designed to extract the base URL from the full URL of the target page. For example, if the full URL is http://domain.com/article, the base URL extracted will be http://domain.com.

3) **Fetching robots.txt:**

- Once we have the base URL, we construct the URL to the robots.txt file by appending/robots.txt to the base URL. For example, if the base URL is http://domain.com, the robots.txt URL will be http://domain.com/robots.txt.

- The **Robots.fetch** method is used to fetch and parse the robots.txt file. This method sends a request to the robots.txt URL and retrieves its contents.

- The robots.txt file is a plain text file that specifies the crawling permissions for different parts of the website. It contains directives such as User-agent, Disallow, and Allow, which control the behavior of web crawlers.

  – **User-agent:** This directive specifies which web crawlers the rules apply to. For example, User-agent: * applies to all crawlers.

```
User-agent: *
Allow: /
Disallow: /connector/
Disallow: /plugins.
Disallow: /firststeps.
Disallow: /publicliterature.PublicLiterature.search.html
Disallow: /lite.publication.PublicationRequestFulltextPromo.requestFulltext.html
Disallow: /amp/authorize
Allow: /signup.SignUp.html
Disallow: /signup.
```

**Figure 1  Robots.txt sample file.**                               

    – **Disallow:** This directive specifies the parts of the website that are not allowed to be crawled. For example, Disallow: /private means that the /private directory should not be accessed by crawlers.

    – **Allow:** This directive specifies the parts of the website that are allowed to be crawled. It is typically used to override a Disallow rule. For example, Allow: /public means that the /public directory can be accessed by crawlers even if a broader Disallow rule is in place.

4) **Checking permissions:** The **robots.allowed** method checks whether the specific URL (*e.g.*, http://domain.com/article) is allowed to be accessed by our bot. This check is based on the rules specified in the robots.txt file. The method takes two parameters: the URL to check and the 'User-Agent' (which identifies the bot).

5) **Returning results:** The function returns "allowed" if the URL is permitted for crawling according to the robots.txt file. It returns "disallowed" if crawling the URL is not permitted.

This algorithm is always run before any scraping operations to ensure that we adhere to the legal and ethical guidelines set by the website owners, respecting their preferences as specified in the robots.txt file.

## Scraper bots

If a successful result is returned from the Robots algorithm, this indicates that the page can be downloaded and scraping can be started. Although a basic algorithm is used to scrape pages from different academic databases, different tuning needs have emerged for each site. The different development methodologies of the sites and the different permissions given by the site administrators to the user during page visits have been a factor in this issue. Among these algorithms, the "Requests" library is used to download the page and control the session. The library provides the opportunity to perform operations on the page by managing cookies and providing authentication where necessary while visiting the page (https://pypi.org/project/requests/; *Python Software Foundation, 2023d*). The popular BeautifulSoup scraping library was used to enable the bot to scrape. This library takes the downloaded page as a source and transforms it into an HTML tree. In this HTML tree, the relevant element of the page can be retrieved using methods such as find by element type, find by class, find by xpath (https://pypi.org/project/beautifulsoup4/; *Python Software*

*Foundation, 2023a*). For pages with a different technical development methodology and where the HTML tree cannot be extracted, the article summary was obtained by accessing the page metadata. For this purpose, the "extraction" library was used to obtain the downloaded page metadata (https://pypi.org/project/extraction; *Python Software Foundation, 2023b*). We can summarise the common structure designed to scrap data from various academic sites (Springer, ACM, IEEE and ResearchGate) as follows.

1) **Page download:**

- Use the **Requests** library to download the web page and manage sessions, cookies, and authentication.

2) **HTML parsing:**

- Utilize the **BeautifulSoup** library to transform the downloaded page into an HTML tree.
- Extract relevant elements using methods like **find by element type**, **find by class**, and **find by xpath**.

3) **Metadata extraction:**

- For pages where the HTML tree cannot be directly extracted, use the **extraction** library to access the page metadata and obtain the article summary.

### Algorithm for Springer & ACM

It was sufficient to use the common algorithm to scrap the data from ACM and Springer academic databases. This algorithm extracts the HTML tree of the page with BeautifulSoup after obtaining the cookies of the page with the "Requests" library and downloading the page. According to this tree, the HTML element containing the article summary section is obtained and the article summary is reached (http://link.springer.com, *Springer Nature, 2023*; http://dl.acm.org/, *Association for Computing Machinery, 2023*).

### Algorithm for IEEE

As the IEEE academic research website was developed using a different methodology to others, we found that the HTML tree of the site could not be downloaded using the Requests library. Thinking that this site, which is widely used for literature searches, would be useful for our application, we looked for different ways to access the article abstracts. By examining the article pages of the site, we found that the article summary was also included in the page metadata. And we targeted the metadata for scraping. We used the extraction library to extract the page metadata and access the article summary (https://ieeexplore.ieee.org; *IEEE xPlore, 2023*). The underlying change for the IEEE site is that the HTML tree cannot be downloaded using the Requests library, so the algorithm accesses the article abstract from the page metadata using the extraction library.

### Algorithm for researchgate

The common algorithm used on the Springer and ACM sites also works on the ResearchGate site. However, in tests where the number of pages was more than 100, the site stopped our scraping bot with a warning that "please login to make page requests this often". To overcome this problem, we looked at the network exchange of the page during login, and constructed this exchange within the algorithm using the Session class in the Requests library. Accordingly, the algorithm logs in before sending the request, as the site warns us, and then scrapes the article summary from the HTML tree using the BeautifulSoup library (https://www.researchgate.net/; *Research Gate, 2023*). The scrap process for the Researhgate site is similar to Springer and ACM, but with the addition of a login step that uses the Session class in the Requests library to handle frequent page requests.

## Backend structure

Once our algorithms had finished searching article sites for the specified keyword and extracting article summaries from the URLs that were found, we developed backend features to structure these methods. To store the keywords that were searched, the URLs that were found, the research done on these URLs, and the data gathered over the course of the application lifetime, a cloud-based database has been established. A service has been created that will reply to all requests made *via* the interface by combining algorithms with this database. Figure 2 shows the application flow together with the backend architecture of the program.

### Database layer

A database was created to store all transaction logs from the application, the details of the search, the URLs found as a result of the search, and the article information extracted from the URLs found. The unstructured and cloud-based Cloud Firestore database provided by Google was used for our high-volume application. It was preferred because its access is free and flexible in the cloud environment. The database diagram is shown in Fig. 3.

### Service layer

A web service was developed for the application to provide database access to the interface and to respond to user requests. Our service, which takes on the task of completing the crawling and scraping processes and quickly sending the results to the user, has been developed using object-oriented principles in the Python language in order to be compatible with the crawling and scraping algorithms developed previously. Within the service, there are APIs that serve different purposes for all requests coming from the interface. Auth API for authentication requests, Crawler API for starting crawling operations and writing to the database, Scraping API for scraping on crawled URLs and writing article details to the database have been developed. In order to respond quickly to user requests, performance is of paramount importance and all processes are multi-threaded. To prevent long running processes from affecting the user, the processes are executed in the background using background tasks and when the process is complete, the user is notified using mail/in-app notification methods. The open source FAST API library

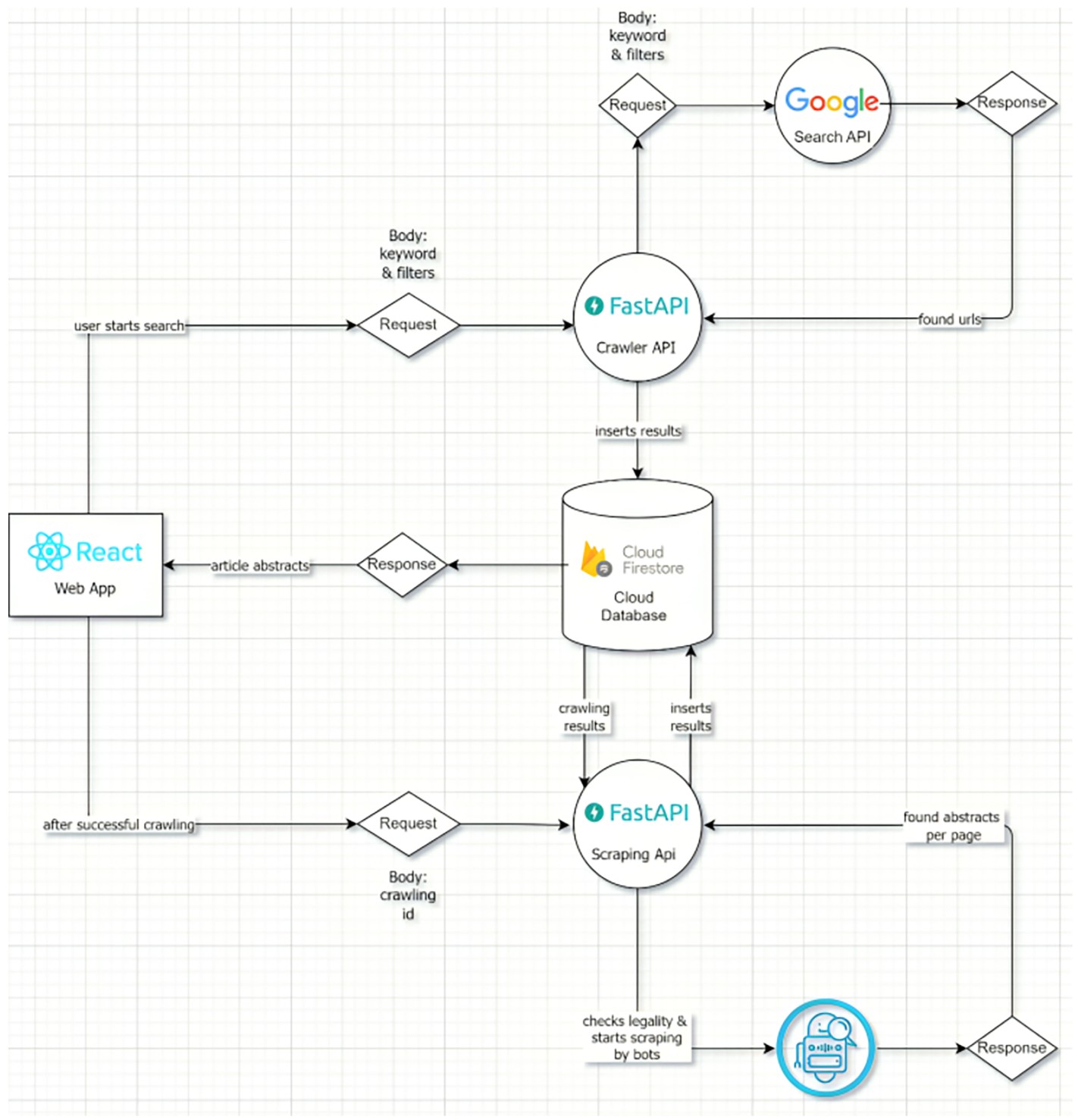

**Figure 2 Backend structure of app.**

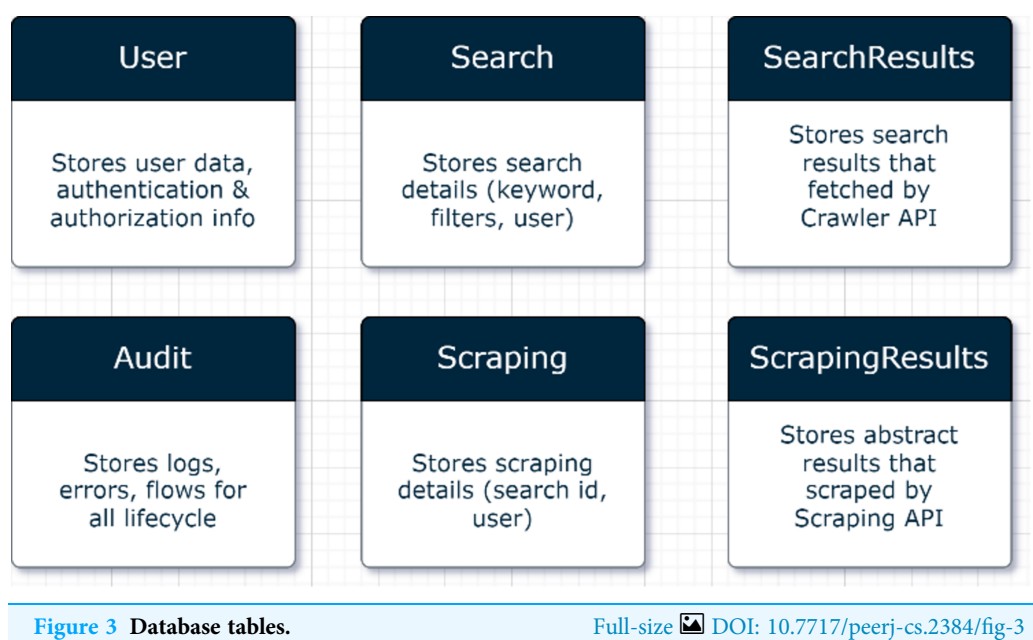

**Figure 3 Database tables.**                               

was used to develop this web service due to its popularity in recent years, its extensive documentation and low development effort. This open source framework, which was first developed in 2018, has replaced Flask, which was used for the same purpose in previous years, due to its speed and ease of use by the application development boilerplate.It is an open source, language-independent library that can be considered as the documentation of a backend service, providing information on how requests should be sent to endpoints within the service and enabling testing. In today's applications, instead of writing a usage document for backend operations, it is preferable to create Swagger documentation that can both enable testing and describe usage details. Swagger documentation is automatically generated by the FAST API library that we utilize. The "api_address: port/docs" is used to serve the generated api endpoints once it has automatically identified the Python schemas utilized. Figure 4 shows the swagger documentation for a portion of our application. The open-source Pydantic package is used to define class object structures, configure them, and produce useful Python schemas—the building blocks of object-oriented programming. It has been favored because it simplifies and makes Python schema objects more understandable while being compatible with both Swinger and FAST API. The request and response objects in our web service are defined using Pydantic. This provides convenience in terms of code readability as well as swagger documentation.

### Application

We have developed a user-friendly interface, as shown in Fig. 5, so that the user can perform all crawling and scraping operations safely and quickly. As shown in the interface of the application in Fig. 5, the researcher can filter the date in addition to the keywords he/she wants to search. In addition, the researcher can filter the words he/she does not want to include by entering them in the "Excluded Words" field. By adding words to the 'Exact Words' field, it is also possible to filter for words that should be found exactly in the search

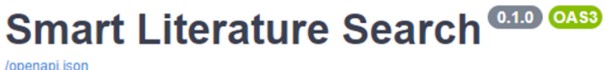

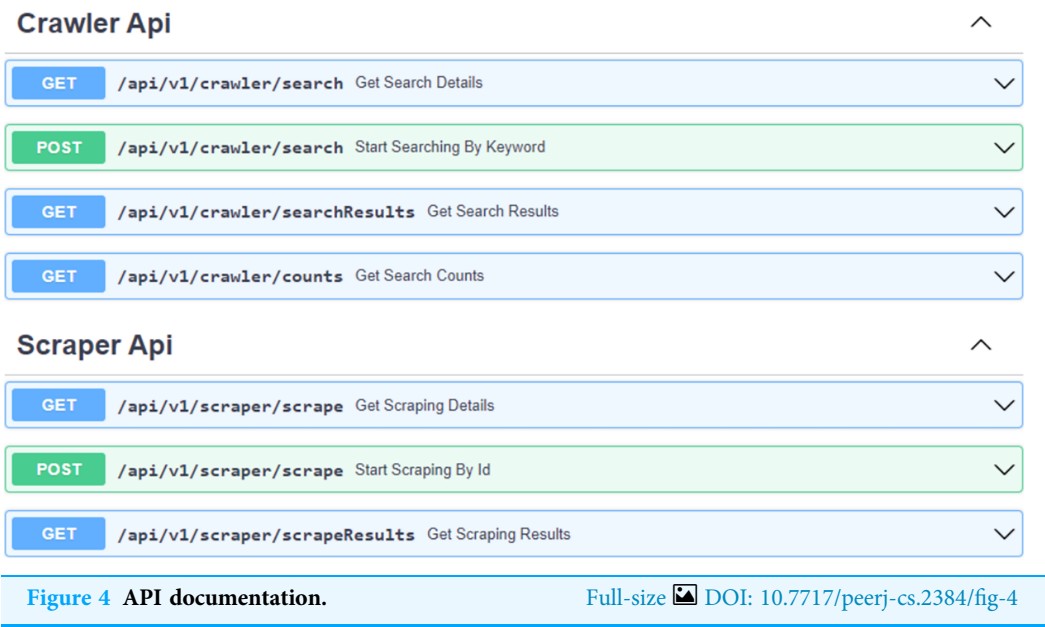

**Figure 4  API documentation.**

criteria. Studies are collected from relevant academic databases according to the criteria entered by the user *via* the interface. These collected studies are ranked according to various criteria when presented to the user. The following four criteria are taken into account in the ranking process.

- **Keyword relevance:** Abstracts are selected based on their match to user-defined keywords, ensuring that only those with high relevance to the research topic are retrieved.

We use the Term Frequency-Inverse Document Frequency (TF-IDF) algorithm to evaluate the importance of keywords in each document. The TF-IDF score helps in ranking documents based on the relevance of the keywords provided by the user. TF-IDF algorithm is a widely used method to evaluate the importance of a word in a document relative to a collection of documents (*corpus*). This method helps in identifying the most relevant keywords in a document by considering both the frequency of the term in the document and its occurrence across the *corpus*.

*Term Frequency (TF):* Term frequency measures how frequently a term appears in a document. The assumption is that the more a term appears in a document, the more important it is within that document. The term frequency for a term **t** in a document d is calculated as:

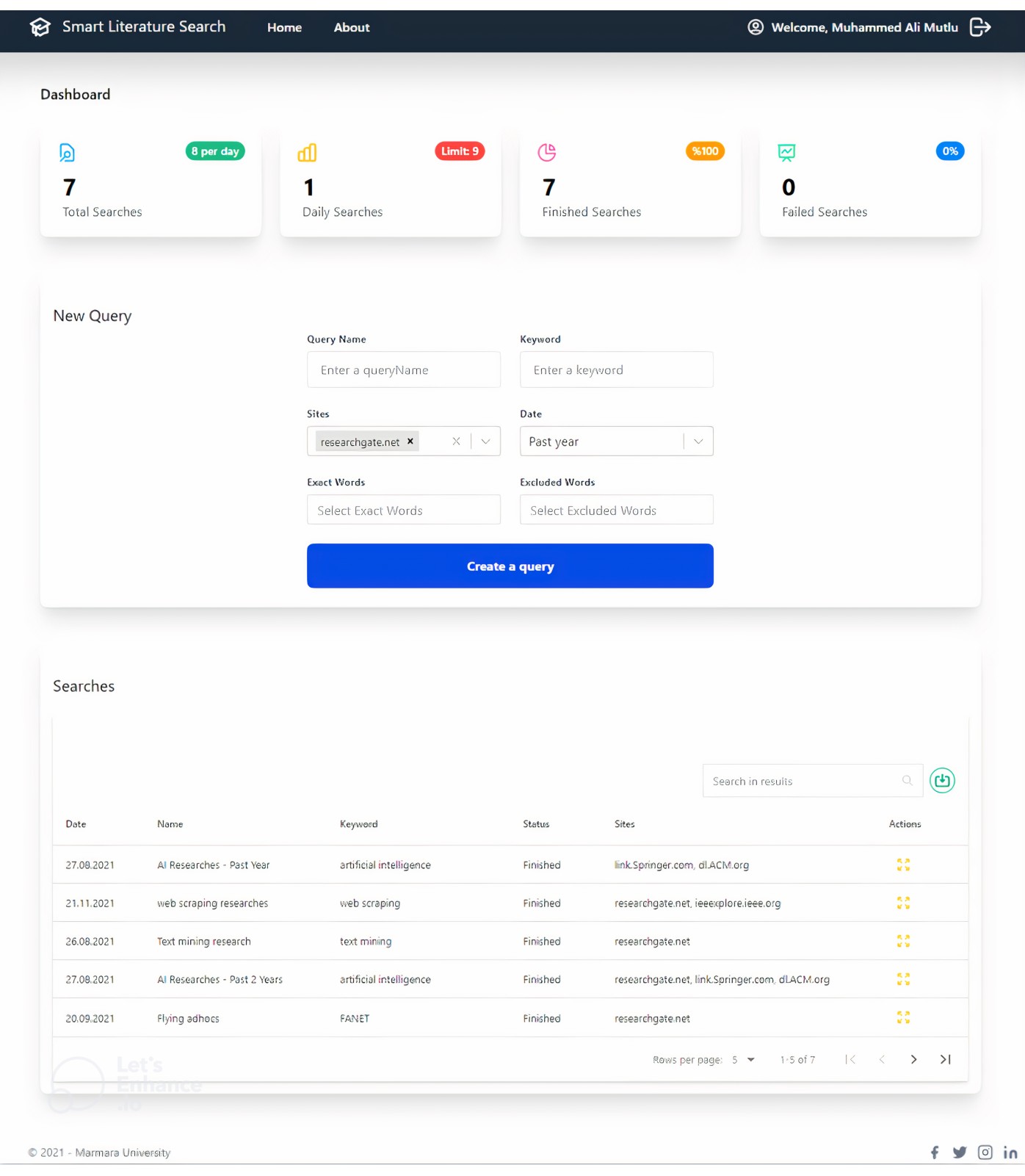

**Figure 5** **Interface of the application.**

$$TF(t, d) = f(t, d)/N_d$$

– $f(t, d)$ is the number of times term **t** appears in document **d**.

– $N_d$ is the total number of terms in document **d**.

*Inverse Document Frequency (IDF)*: Inverse Document Frequency measures the importance of a term across the entire *corpus*. The idea is that terms that occur in many documents are less informative than those that occur in fewer documents. The inverse document frequency for a term **t** is calculated as:

$$IDF(t, D) = log\left(\frac{N}{|d\varepsilon D : t\varepsilon d|}\right)$$

– **N** is the total number of documents in the *corpus*.

– $|d\varepsilon D : t\varepsilon d|$ is the number of documents in which term **t** appears.

*TF-IDF calculation*: The TF-IDF score for a term **t** in a document **d** is the product of its TF and IDF scores:

$$TF - IDF(t, d, D) = TF(t, d) x IDF(t, D)$$

This score reflects the importance of the term **t** in the document **d**, adjusted for how commonly it occurs in the *corpus* **D**. How the keyword relevance process is performed is shown in the flowchart in Fig. 5.

- **Publication date:** In order to ensure that users focus on current publications, the most recent publications are retrieved according to the specified range.

We parse and sort the publication dates of documents to ensure that the most recent publications are prioritized. To ensure that the most recent publications are prioritized in our search results, we employ a process to parse and sort the publication dates of documents. This involves extracting the publication dates from the metadata of each document, converting these dates into a standard format, and then sorting the documents based on these dates.

To ensure that the most recent publications are prioritized in our search results, we follow a process involving the extraction, parsing, and sorting of publication dates. Initially, we extract the publication dates from the metadata of each document, which may come in various formats. These dates are then parsed and converted into a standard format, such as **YYYY-MM-DD**, using Python's **datetime** library, ensuring consistency across all documents. Once standardized, the documents are sorted in descending order based on their parsed publication dates. This sorting process uses the **sorted** function with the parsed date as the key, which ensures that the most recent publications appear first in the search results. This method enhances the relevance and timeliness of the information provided to the user, making sure that they have access to the latest research developments. The flowchart of the algorithm used to obtain the publication date information of the articles is shown in Fig. 6.

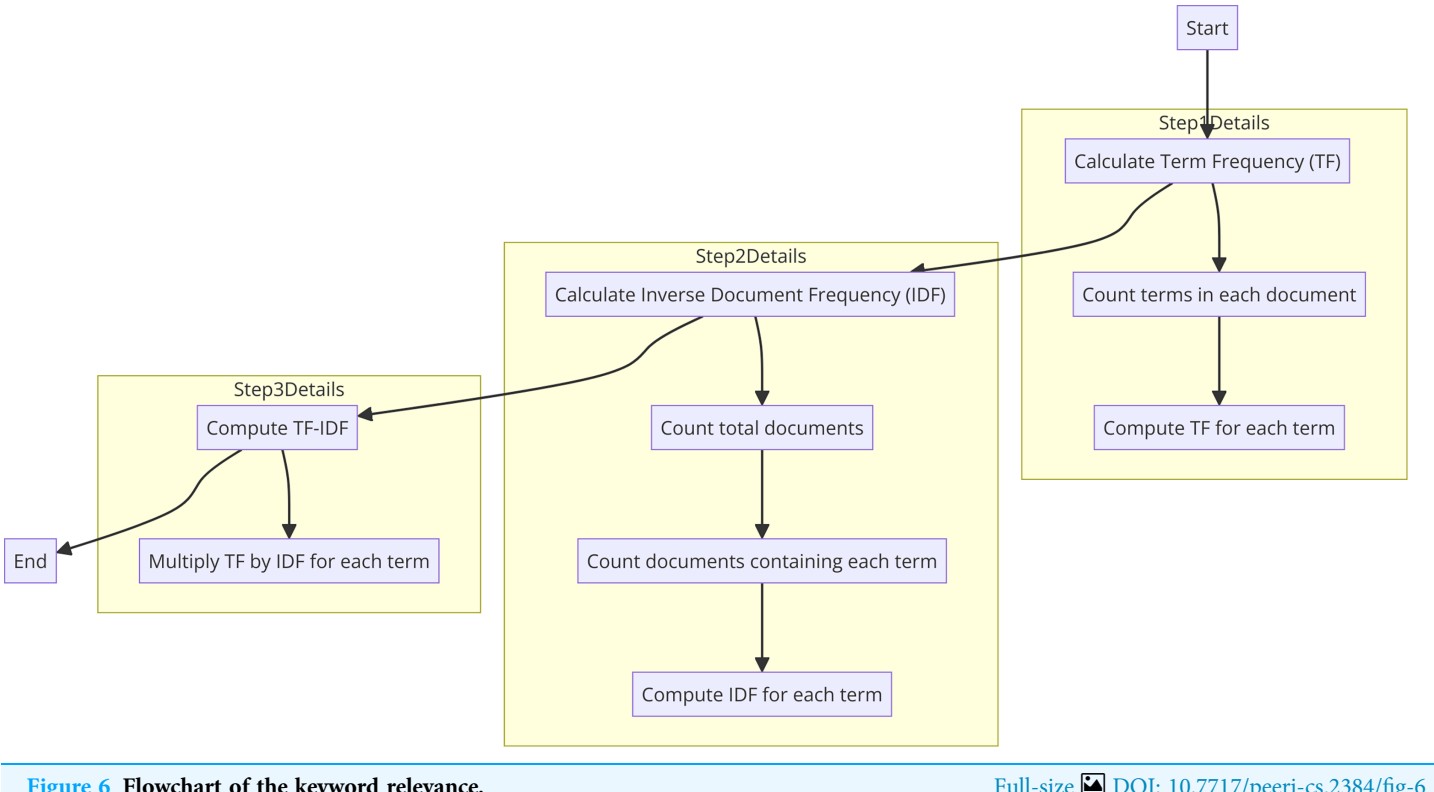

**Figure 6 Flowchart of the keyword relevance.**

By parsing and sorting publication dates, we ensure that our search results prioritize the most recent research, which is crucial for staying up-to-date with the latest developments in the field. This method enhances the relevance and timeliness of the information presented to the user, making sure that they have access to the latest research developments.

- **Citation analysis:** The application evaluates the number of citations as an indicator of the impact and relevance of the abstract and presents the studies that make a significant contribution to the field to the user first.

In our study, citation analysis is used to evaluate the impact and relevance of academic documents. The number of citations a document has received is a strong indicator of its influence within the academic community. Documents with higher citation counts are considered to have made significant contributions to their field, and therefore, they are prioritized in our search results.

In our application, citation analysis involves retrieving the citation counts for each document from academic databases such as Google Scholar, Scopus, or Web of Science, which provide metadata including the number of times each document has been cited. These citation counts are extracted and stored along with other metadata. The documents are then ranked in descending order based on their citation counts, using a sorting algorithm that prioritizes documents with higher citation counts. This ranking ensures

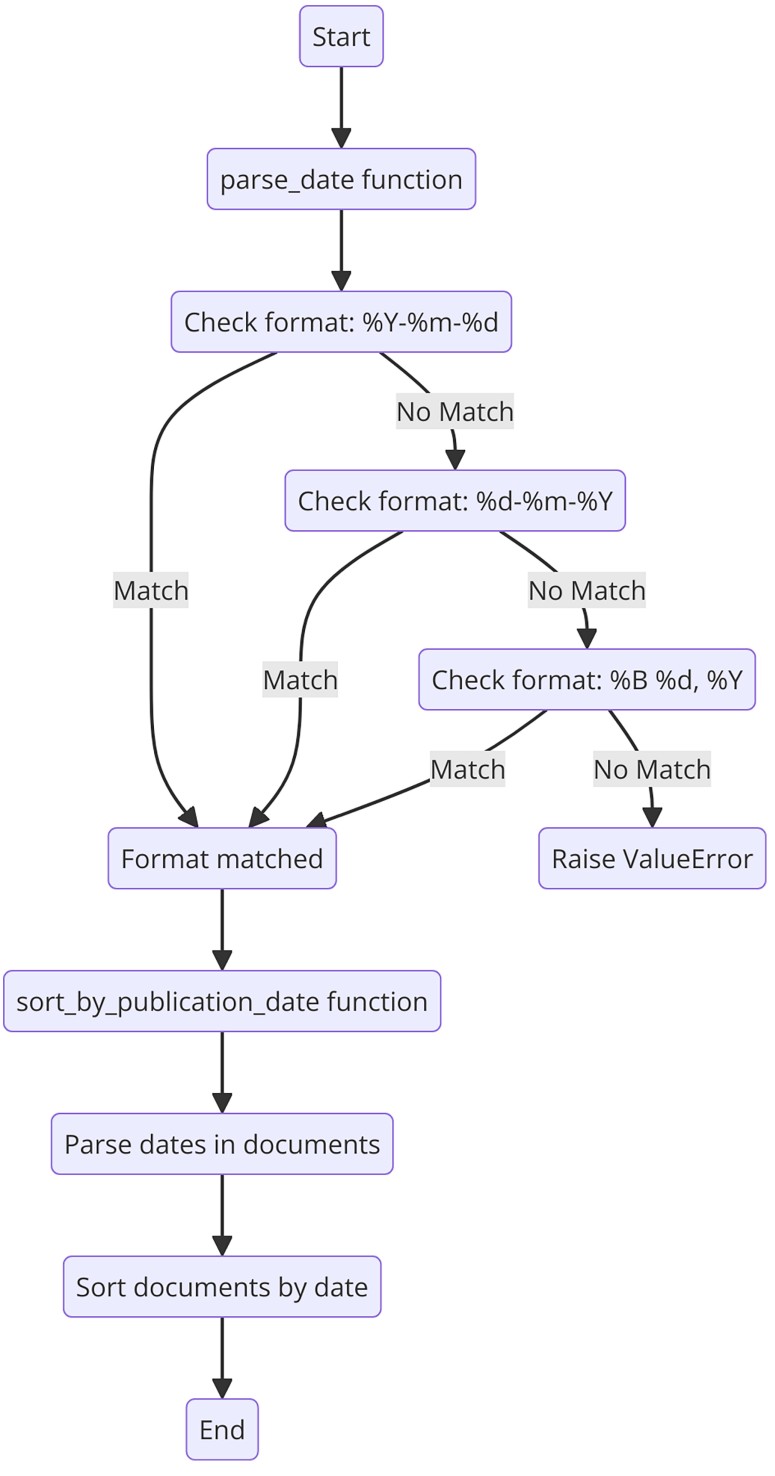

Figure 7 Flowchart of the publication date.

that the most highly cited and influential documents are presented to the user first, highlighting studies that have made significant contributions to their field and thereby enhancing the relevance and impact of the search results. The pseudocode of the approach used for obtaining and ranking the citation numbers is presented in Fig. 7.

- **Journal ranking and Impact Factor:** Abstracts from highly ranked journals, which are more likely to be of high quality and relevance, are prioritized.

In our application, journal ranking is used to assess the quality and influence of the publications within each journal. One of the primary metrics used for this purpose is the Impact Factor, which measures the average number of citations received by articles published in a journal during a specific period. Higher impact factors indicate more influential journals, and articles from these journals are considered more credible and significant.

Our application employs a multi-criteria ranking algorithm to present academic studies to users, ensuring that the studies are ranked by their relevance, recency, impact, and quality. The four criteria used are keyword relevance, publication date, citation analysis, and journal ranking and Impact Factor. Each criterion is assigned a specific weight to calculate the overall score for each document, as follows: keyword relevance (50%), publication date (20%), citation analysis (15%), and Journal Ranking (15%). The final ranking of documents is determined by combining the scores from all four criteria. Each criterion is normalized to a common scale and then weighted according to its specified importance. The overall score for each document is calculated as a weighted sum of the normalized scores from all criteria. The documents are then sorted based on their overall scores and presented to the user. How the publications are ranked and presented in the developed application is shown in the pseudocode in Fig. 8.

The user can quickly communicate with the service *via* the interface shown in Fig. 9 and can select search filters, initiate a search to the backend, view the status of the initiated request and view the data obtained as a result of the search. To develop the interface, we used the React library developed by Facebook and recently used in web development projects. We developed our application as a single page application in order to keep up with today's trending technology standards. In addition to this library, which has a very large community and can be developed quickly and easily with good documentation, we also benefited from the Redux state management library for the application's data management logic. Thanks to this library, we were able to ensure secure communication between the data obtained from the login screen, which is the first screen the user accesses, and the data on the end screen.

## RESULTS AND DISCUSSIONS

### Usage of application

The operations that the user can perform with the application are shown sequentially in the flowchart in Fig. 10. When the application is launched, the user is first asked to log in to the system. If the user does not have a membership in the system, he/she can register to the system by email or by linking his/her Google account. After successfully logging in to the

1. **START**: Begin the process.

2. **Input**: Retrieve the list of documents with metadata.

3. **Step 1**: Initialize an empty list to store citation counts.

4. **For each document**:

   - Extract the citation count from the document's metadata.

   - Append the citation count to the list.

5. **Continue until all counts are retrieved.**

6. **Step 2**:

   - Sort the documents by citation counts.

   - Use the `sorted` function with `key=getCitationCount` and `reverse=True`.

7. **Continue until all documents are sorted.**

8. **Output**: Output the sorted list of documents.

9. **END**: End the process.

**Figure 8  Pseudocode of the citation counts.**

system, the user is taken to the main screen where he/she can use the application. The settings page can be used to update and view personal information. Once the user reaches Google's restricted search limit, they will no longer be able to search for that day. The user can check their remaining search limit from the application's settings page. The user can see the searches they have started, stopped or are still scraping on the Dashboard screen. This page shows the current status of the scraping and information such as when it was searched and what filters were used. The article summaries obtained as a result of the completed searches can be accessed from this page and exported if required. He/she can start a new search by copying the filter and other configurations from a previous search, or by selecting the filters from the Filter menu from scratch, entering his/her keyword, and then selecting the academic websites to be searched.

After the web crawler collects a large volume of data from various academic websites, the information undergoes a multi-step filtering process to ensure that users receive the most relevant and optimal search results. Initially, the collected URLs are stored in a database along with the associated metadata. The filtering process begins by checking the legality of accessing each URL, ensuring compliance with the site's robots.txt file to avoid any legal issues.

1. **Start:** Begin the process with a list of documents and their metadata.

2. **Step 1: Calculate Keyword Relevance:** Compute the TF-IDF scores for each document.

3. **Step 2: Normalize and Weight Keyword Relevance:** Normalize the TF-IDF scores and apply a weight of 50%.

4. **Step 3: Extract and Parse Publication Dates:** Extract and standardize the publication dates from the metadata.

5. **Step 4: Normalize and Weight Publication Dates:** Normalize the publication dates and apply a weight of 20%.

6. **Step 5: Retrieve Citation Counts:** Retrieve the citation counts from academic databases.

7. **Step 6: Normalize and Weight Citation Counts:** Normalize the citation counts and apply a weight of 15%.

8. **Step 7: Assign Journal Impact Factors:** Retrieve and assign the impact factors for the journals in which the documents are published.

9. **Step 8: Normalize and Weight Journal Impact Factors:** Normalize the impact factors and apply a weight of 15%.

10. **Step 9: Calculate Overall Scores:** Compute the overall score for each document by combining the normalized and weighted scores from all criteria.

11. **Step 10: Sort Documents by Overall Scores:** Sort the documents based on their overall scores in descending order.

12. **Output:** Present the sorted list of documents to the user.

13. **End:** The process concludes with the presentation of the sorted documents.

**Figure 9  Pseudocode of the general flow.**

Next, the data is subjected to both Exact Match and Pattern Matching approaches. The Exact Match approach uses built-in string operations and the re library to identify and retrieve documents that precisely match the user's specified keywords. This step ensures high precision by excluding irrelevant documents that do not contain the exact keywords. The Pattern Matching approach is then applied using the re library to compile and search with regular expressions. This allows the system to recognize variations of the keywords, including synonyms and related terms, thus enhancing recall by capturing a broader set of relevant documents.

Further filtering involves analyzing the relevance and quality of the retrieved documents based on several criteria: keyword relevance, where abstracts are evaluated for their relevance to the user's specified keywords; publication date, prioritizing the most recent publications to ensure up-to-date information; citation analysis, ranking articles based on their citation count to indicate their impact and significance in the field; and journal

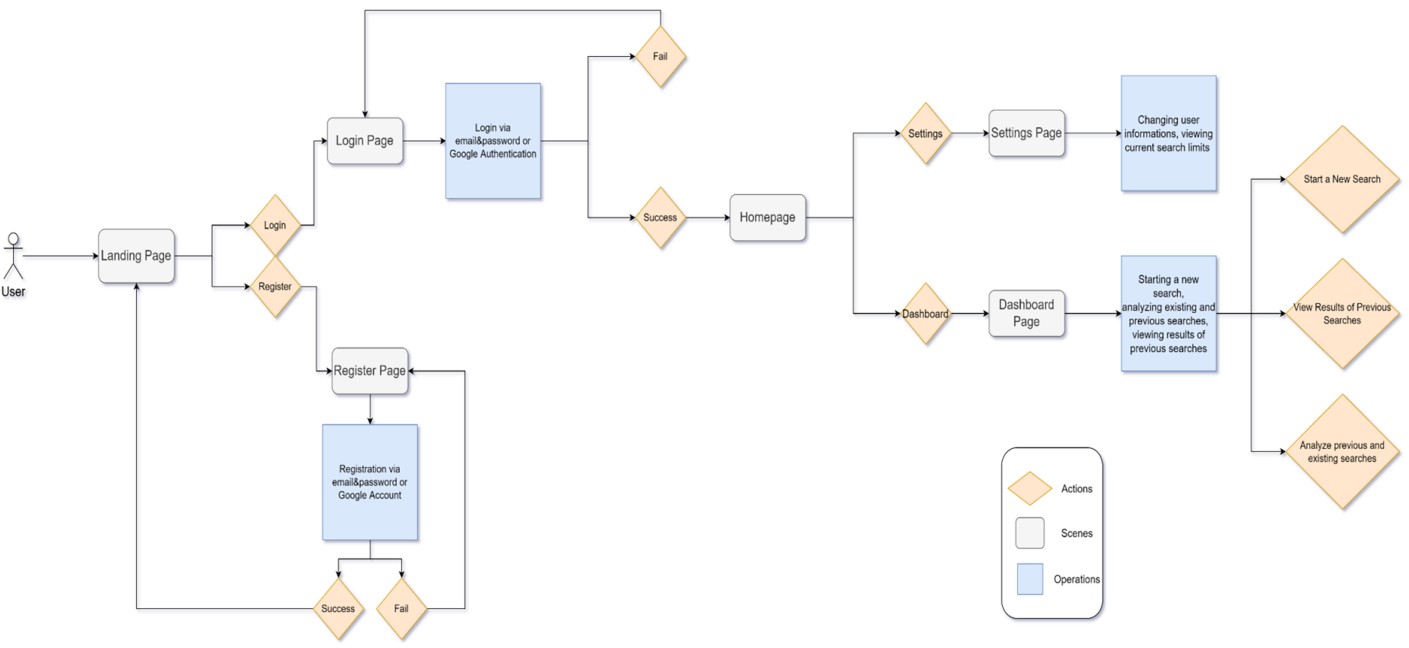

**Figure 10 Sequential flowchart of user operations in the application.**               

ranking and Impact Factor, prioritizing documents from highly ranked journals, as these are typically considered more credible and influential.

The filtered results are then presented to the user through the application's interface, allowing them to quickly and efficiently access the most relevant studies. This comprehensive filtering process ensures that users receive high-quality, precise, and relevant search results, significantly enhancing the efficiency and effectiveness of their literature review process.

## Logic of application

When it is desired to start a new search in the application used with user interaction, the processes that take place are shown sequentially in Fig. 11. After logging into the system, the user selects filters and academic websites and starts a search with the desired keyword. After the crawling operations are completed in the backend service, the results are written to the database. In the next step, scraping operations are performed in the backend service using the crawling ID of the search performed. The results obtained by the scraping service are written to the database. In the last part, the results obtained with the frontend-backend connection are presented to the user.

### Logic of crawling operations

After the user has selected filters and academic sites from the system, he/she starts his/her search. Crawling operations start with a request sent to the crawler API on the service side. The flow of crawling operations is shown in Fig. 12. During the crawling phase, our service first intercepts the user's request and acquires the filters and sites to be crawled according

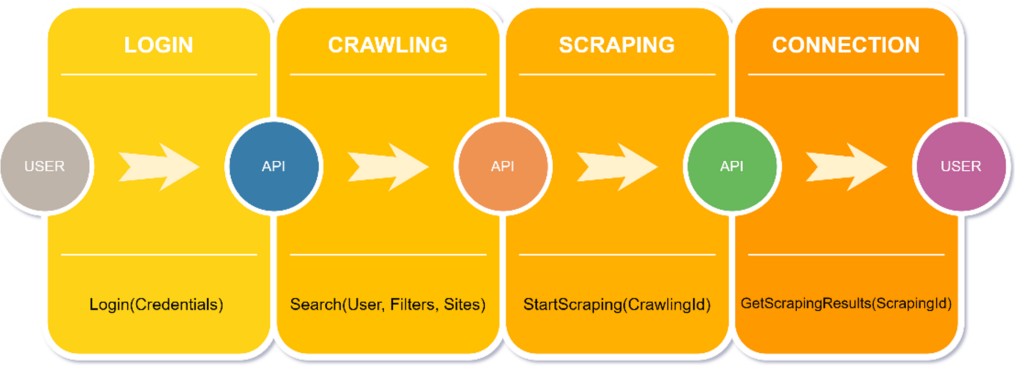

**Figure 11** **Logic of application.**

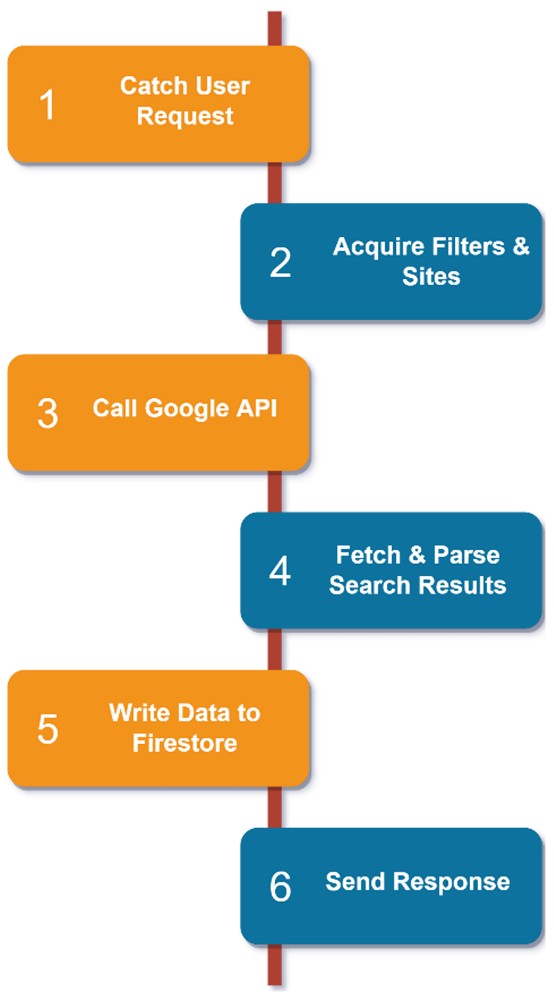

**Figure 12** **Crawling logic.**

to the request schema received. It then makes a call to our custom search engine built with Google using these filters. The URL and title of the sites we need are parsed by accessing the responses provided by the Google Search API, including the large and unnecessary data. These final results are written to our database in Cloud Firestore in parallel with the user information. When the process is complete, a successful response containing the crawling ID is sent to the frontend.

### Logic of scraping operations

The flow of the scraping process in the application is shown in Fig. 13. First, after the successful message is sent to the front-end, the scraping request containing the crawling ID is intercepted by our scraper API on the service side. The crawling id is obtained from the request and the user of that crawl is compared with the requesting user. If the crawling id matches the user, the website URLs associated with that crawl are retrieved from the database. Legality is checked by browsing each URL and checking the robots.txt. Once the legal URLs are obtained, scraping bots are run on the web pages at those URLs. Then abstracts are obtained with appropriate algorithms by visiting each legal article page. These abstracts are then written to our Cloud Firestore database. When the process is complete, a successful response containing the scraping ID is sent to the front-end.

## Outputs

In the developed "Smart Literature Search" application, a test was conducted with the "web scraping" keyword on Postman. The search was restricted to the "IEEE" and "Researchgate" academic databases. The Crawler API searched the relevant URLs from the specified websites and wrote them to the database within 12 s. The recorded data showed that a total of 100 results were obtained, including 39 contents related to the relevant keyword from the "IEEE" academic website and 61 from the "ResearchGate" academic website. Then, a request was sent for scraping operations using the Scraper API with the relevant search ID. Scraping processes were started as a background task and completed in approximately 5 min. The obtained article abstracts were saved in the database. As a result of the scraping operations, 84 URLs, which are academic articles, were extracted from a total of 100 URLs, and the abstracts of these extracts were recorded in the database. The total time taken to complete the crawling and scraping processes for the searched keyword is approximately 6 min. This elapsed time is done in the background without being felt by the user with the queue structure used in the application, and when the processes are completed, it can be followed from the status area for the relevant search on the dashboard page. It starts with the "Started" status first, if there are other processes, it is taken to the "Queued" status, otherwise it is taken to the "Finished" status after waiting for it to end. After this step, the user can see the results. The process performed in each stage, how many URLs were obtained as a result of this process and how long it took to complete it are summarised in Table 2. In addition, Fig. 14 shows the time spent for data collection, data scraping and overall process for this sample search.

With the developed application, the user obtained the abstract information of 84 articles that meet the criteria he/she determined over two academic databases in 6 min. If he

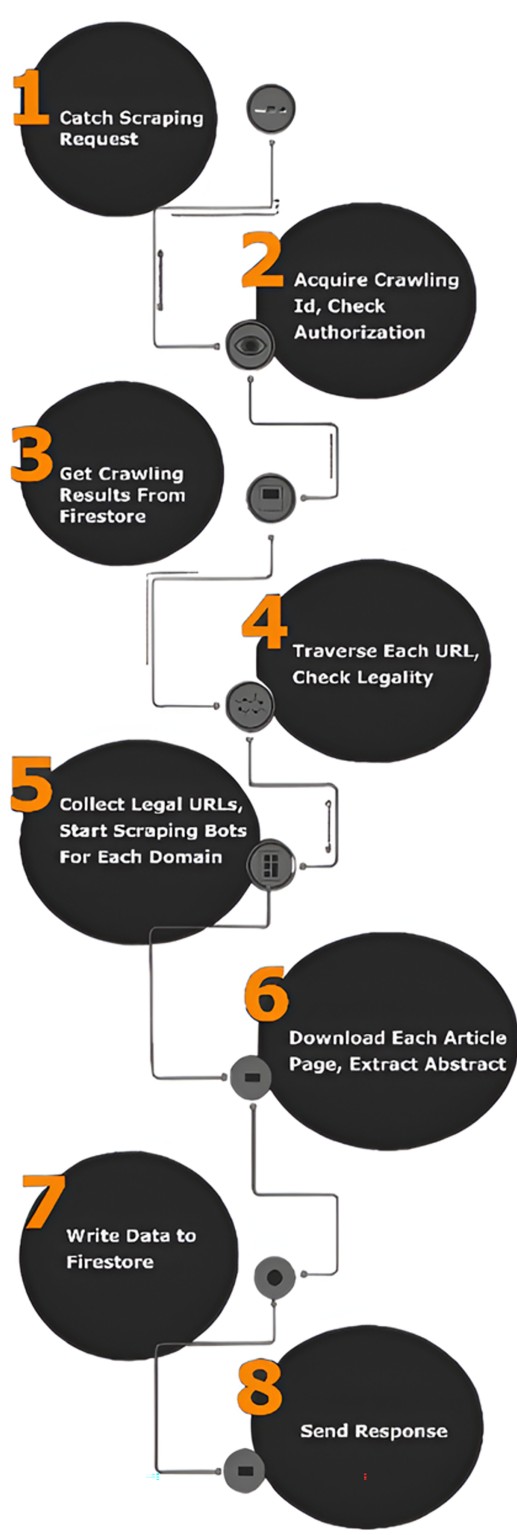

**Figure 13  Scraping logic.**

**Table 2 Performance metrics for each stage.**

| Stage | Metric | Value |
|---|---|---|
| Data collection | Total URLs found | 100 |
| Data collection | IEEE URLs found | 39 |
| Data collection | Researchgate URLs found | 61 |
| Data collection | Time taken (seconds) | 12 |
| Data scraping | Total URLs Scraped | 84 |
| Data scraping | Time taken (minutes) | 5 |
| Overall process | Total time (minutes) | 6 |

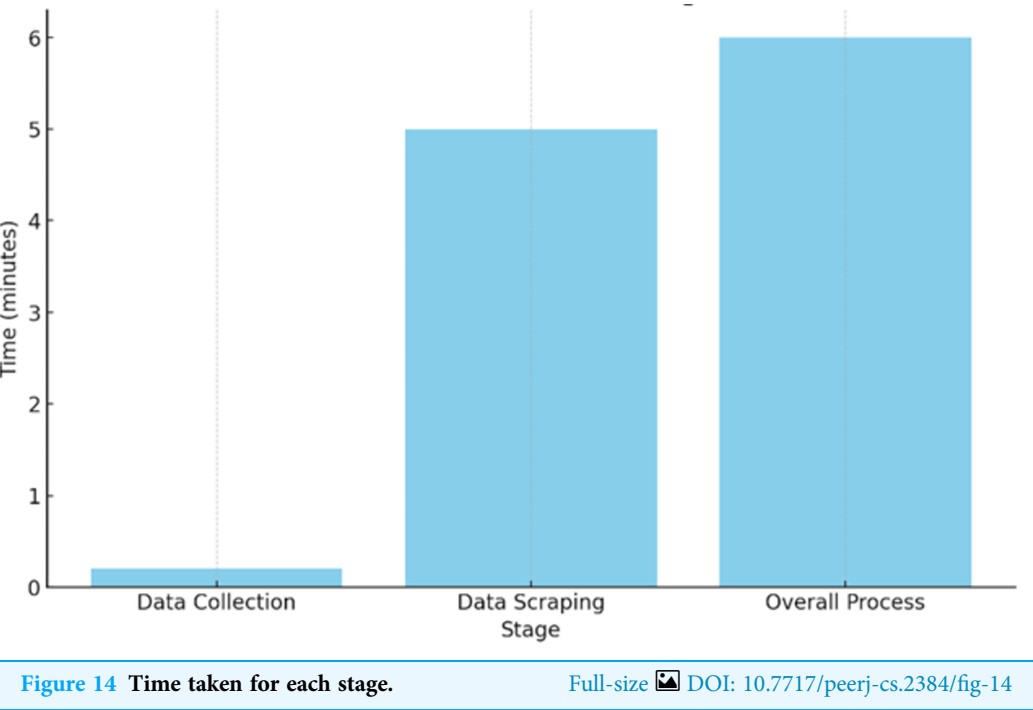

**Figure 14 Time taken for each stage.**

wanted to perform this search manually, he would have to open each article separately by entering the search criteria separately in each database, which would take a very long time.

## Advantages of the developed application

Artificial intelligence-based methods are becoming more and more common these days to help researchers find material more quickly and effectively. There are various benefits associated with using the web scraping/crawling method for text-based applications as opposed to an artificial intelligence (AI) based approach. Compared to artificial intelligence-based applications, the web scraping and web crawling application developed for this study has a number of advantages. These advantages are primarily attributable to its capacity to operate with fewer resources, reduce complexity, and pay attention to legality control. The following is a list of benefits that the developed application has over artificial intelligence-based methods.

**Table 3 Web scraping/web crawling and artificial intelligence-based application.**

| Aspect | Web scraping/web crawling | Artificial intelligence-based application |
|---|---|---|
| Complexity | Simpler to implement and understand. | May involve complex algorithms and data models. |
| Data acquisition | Highly efficient for direct data collection. | Focuses on interpreting or understanding text, which can be less direct. |
| Computational resources | Requires lower computational resources. | Often requires significan generalized and less predictable. |
| Adaptability to data format changes | Easier to adapt scrapers/crawlers to website changes. | Adjusting AI models to new data formats can be more complex. |
| Transparency | Transparent data collection process. | The data processing logic can be opaque, making it hard to trace data sources. |
| Deployment speed for specific tasks | Quick deployment for targeted data collection tasks. | Requires data collection, training, and validation phases. |
| Suitability | Best for applications where the primary goal is data collection. | Best for applications requiring data interpretation, analysis, or generation. |

- **Simplicity and accessibility:** Web scraping and web crawling techniques are generally simpler to implement and understand than AI-based methods. They do not require extensive datasets for training or complex algorithms, making them more accessible to developers with different levels of expertise.

- **Efficiency in data collection:** Web scraping and crawling are highly efficient for collecting data directly from web pages. They can quickly collect textual data, such as abstracts from academic articles or content from websites, without the need to interpret or understand the text, which is important in applications where the main goal is to collect data.

- **Lower computational resources:** Unlike AI applications that can require significant computing power for processing and analysis (especially for training machine learning models), web scraping and crawling can be executed with relatively low computational resources. This makes them more suitable for applications with budget constraints or where computational efficiency is a priority.

- **Direct control over data source and structure:** With web scraping and crawling, developers have direct control over which web pages are targeted and how the data is extracted. This allows precise customization of the data collection process to suit specific needs. This provides a significant advantage over AI methods, which can require larger datasets and sometimes produce unpredictable outputs.

- **Easier to adapt to changes in data format:** When a website changes its layout or structure, updating a web scraper or crawler to accommodate these changes can be easier than retraining an AI model, which may require collecting new training data and adjusting the model architecture.

- **textbf transparency in data collection:** The web scraping and crawling process is transparent, which means it is easier to keep track of what data was extracted from where. This is especially important in academic research or applications where the data

source needs to be specified or verified. On the other hand, AI-based text analytics may not always provide clear insights into how data is interpreted or extracted.

- **Quick application for specific tasks:** For specific tasks, such as collecting all summaries containing specific keywords from a set of web pages, the web scraping or crawling approach can be applied quickly and efficiently without the data training stages required by AI applications.

Web scraping and Web crawling based applications and artificial intelligence based applications are presented in Table 3 comparatively within the scope of these features.

## CONCLUSION

The most crucial and initial step in academic studies, the literature evaluation, also establishes the direction of future research. The act of conducting a literature review enables us to locate previous research on topics linked to our planned study and, consequently, to pinpoint opportunities for potential new research. Using web scraping techniques, a system called "Smart Literature Search" has been created for this study in order to expedite and simplify the literature review process. If a researcher is looking for a specific article or is familiar with the literature in the topic, they may forgo the normal approach of reviewing general search engine results. Making this strategic decision can help you save time, reach more articles that are directly relevant to you based on recommendations or citations, enhance the quality and relevancy of your content, and give you access to full texts. We have created a user-friendly application that can retrieve abstracts of publications from popular scholarly research websites, giving us access to a large number of studies. This program allows the user to log in and do new searches in addition to viewing previously conducted searches and results. When using normal methods through browsers or article sites, literature search can take weeks to complete. The developed program makes it possible to complete this task considerably more quickly. Furthermore, when conventional procedures are followed, the application built gives the user access to content that they might otherwise overlook but find useful. This increases the literature review's effectiveness and scope. Academicians and graduate students in our department have tested and begun using our program, which can be searched using a search engine that may be customized by the user as needed. The established system makes it possible to access the vast majority of previously published studies in the topic under investigation, and it streamlines and regularizes the process of conducting a literature review. Web scraping is a very popular technology to get some information from the web. In the literature there are few examples of academic research being scraped. They did this kind of study for analysis on the sites. There is no example for dynamic scraping of academic research sites by user keywords. But we give a solution, an application to scrape academic research by user's own. The user can scrape academic sites at any time with his own keywords by using our "Smart Literature Search App". The developed application allows users to conduct literature searches quickly and easily. While our work provides valuable insights into the development and implementation of a web scraping tool for literature reviews, it is important to recognise its limitations. A major limitation

encountered is that we cannot access certain databases due to permissions specified in robot.txt files, which limits our web scraping capabilities. This limitation may result in a selection bias as some relevant studies that are behind paywalls or have restricted access cannot be included. However, with new features to be added to this system in future work, literature searches will become much more efficient. One of the features that will be added is the ability for users to automatically search at specific time intervals to keep up to date with their previous research topics. In this way, users will be able to browse newly published articles related to their areas of work. In our application, we meet the user directly with the search results generated by keywords and filters. At this stage, no operation is performed on our structurally distributed data. The future aim of the work is to obtain the most accurate and relevant results by using text mining algorithms, in order to be more useful to the user and to make the data more meaningful. As a study that can be carried out on structured and meaningful data, we want to present analytical results and predictions. In this context, we want to present the results obtained in the light of the user's previous searches with images and suggest keywords related to the searches made. Thanks to this suggestion system to be developed, we will try to ensure that the researcher using our application will be able to establish a link between the fields in which he works and the trend topics. To improve performance and ease of use, a trend search scenario will be implemented. If someone started with a keyword that the current user wants to start with, the application will suggest to explore these results in the database. The last feature to be added is suggestions. Text mining algorithms will be used to offer suggestions to users related to their keywords. For example, if a user searches for electric cars *via* Smart Literature Search, the app will suggest searching for Tesla, Tesla technology, *etc.* by looking at and analysing older searches in the database.

## Future trends in web scraping for literature research

The field of web scraping for literature research is continuously evolving, driven by advancements in technology and the increasing demand for efficient data collection methods. This section discusses potential future trends and technological advances that could further enhance the effectiveness and efficiency of web scraping applications.

Future web scraping applications are expected to leverage more advanced AI techniques, such as machine learning and natural language processing (NLP), to automate and improve the accuracy of literature searches. AI-powered web scrapers can better interpret complex web pages and extract relevant data more accurately. Additionally, NLP can provide a deeper contextual understanding of search queries, resulting in more relevant and precise search results.

Advancements in UI and UX design will make web scraping tools more accessible and user-friendly for researchers with varying levels of technical expertise. Interactive dashboards can provide intuitive data visualization, and personalized search features can allow users to customize and save their search preferences, improving the overall user experience.

As web scraping technologies advance, it is crucial to adhere to legal and ethical guidelines. Future improvements may include compliance automation tools that ensure

web scraping activities comply with legal requirements and website terms of service. Additionally, implementing ethical scraping practices will help protect privacy and sensitive information.

The integration of advanced technologies and adherence to ethical practices will play a significant role in the future of web scraping for literature research. These advancements have the potential to enhance the efficiency, accuracy, and accessibility of literature reviews, benefiting researchers across diverse academic disciplines.

### Funding
This work was supported by NTT Data Business Solutions Turkey, Istanbul, Turkey. The funders had no role in study design, data collection and analysis, decision to publish, or preparation of the manuscript.

### Grant Disclosures
The following grant information was disclosed by the authors:
NTT Data Business Solutions Turkey, Istanbul, Turkey.

### Competing Interests
Muhammed Ali Mutlu, a graduate of our Master's Program, is employed by NTT DATA Business Solutions. He is currently working as Head of Data Platforms at NTT DATA Business Solutions.

### Author Contributions
- Muhammed Ali Mutlu conceived and designed the experiments, performed the experiments, analyzed the data, performed the computation work, prepared figures and/or tables, authored or reviewed drafts of the article, and approved the final draft.
- Eyup Emre Ulku conceived and designed the experiments, performed the experiments, analyzed the data, authored or reviewed drafts of the article, and approved the final draft.
- Kazim Yildiz analyzed the data, authored or reviewed drafts of the article, and approved the final draft.

### Data Availability
The frontend and backend code is available in the Supplemental Files.

### Supplemental Information
Supplemental information for this article can be found online at http://dx.doi.org/10.7717/peerj-cs.2384#supplemental-information.

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
