# Peer review of "A web scraping app for smart literature search of the keywords"

_PeerJ Computer Science, doi:10.7717/peerj-cs.2384_

## Round 0.1 · original submission · Major Revisions

Dear authors,

You are advised to critically respond to all comments point by point when preparing a new version of the manuscript and while preparing for the rebuttal letter. Please address all the comments/suggestions provided by the reviewers.

Kind regards,
PCoelho

Reviewer 1 ·

Basic reporting

The document highlights the importance of literature review processes and the benefits of using web scraping technologies for efficient research. While it covers the basic concepts of literature review and web scraping, a more detailed analysis of the implementation of web scraping technologies and their impact on research efficiency could enhance the document's depth.
The content is written in clear and technically correct English. The authors effectively discuss the importance of detailed bibliographic research, the use of search mechanisms, and the development of tools to make research processes more efficient. They also explain the relevance of tracking techniques and web scraping in automating bibliographic research. Overall, the text is clear and effectively conveys key concepts related to intelligent bibliographic research methodologies. The authors provide clear and precise descriptions of various studies and technologies related to web scraping and literature review processes. The language used is formal, academic, and appropriate for the subject matter, demonstrating a high level of professionalism in communication.
The article presents a detailed introduction to the importance of literature review and its role in guiding innovative studies. In the “INTRODUCTION” section, the authors emphasize the relevance of a well-conducted literature review for the success of projects, articles, or theses, highlighting the need to thoroughly research a subject and systematically collect data. They discuss the importance of identifying problems and keywords, determining focus areas, and selecting appropriate resources. The text establishes the connection between the literature review process, the formation of innovative studies, and problem-solving methodologies, underlining the need for reliable sources and systematic examination. Additionally, it addresses the development of intelligent literature review applications, such as web scraping, to increase efficiency. The introduction provides a comprehensive context, demonstrating how the research fits into the broader field of knowledge.
The document references prior literature and related works in the areas of literature review processes, web scraping techniques, and research automation. The authors cite various studies in the field, including works by Dwivedi et al. (2021), Sheela and Jayakumar (2019), Sivarajah et al. (2017), Patel (2019a, 2019b), Jin et al. (2020), Pandey et al. (2018), Liang (2020), and Amália et al. (2018), among others. This extensive referencing of previous studies demonstrates a thorough literature review process and contributes to the credibility of the article.

Experimental design

The methods and materials used in this article are adequately described with detailed information for reproduction by another researcher.
In the “MATERIALS AND METHODS” section, the authors explain the development of the application in a modular structure with a user-friendly interface, allowing users to search for keywords, apply filters, and quickly access search results. The application was developed in four phases: development of the web crawler, development of the web scraper, development of the backend, and development of the frontend.
Additionally, detailed steps for the web crawling process are described, including the development of a crawling bot using the Google Search JSON API service. The document also details customizations made to the application’s API, such as exact matching and pattern matching approaches based on user criteria. The article also discusses the advantages of the application over AI-based methods, emphasizing simplicity, accessibility, and efficiency in data collection through web scraping and crawling techniques.
Including a discussion on possible future trends in web scraping applications for literature research and how technological advances can further improve the process could add value to the document. However, overall, the description of methods and materials provides sufficient information for replication by another researcher, making the process clear and reproducible.
Suggested improvements: Address legal and ethical considerations related to web scraping, especially concerning data privacy and copyrights. Explain how the application ensures compliance with legal frameworks and respects the terms of service of the websites scraped.

Validity of the findings

The conclusions are formulated appropriately, covering the main contributions of the developed application to literature review processes. The authors summarize the contributions in four points: Time Efficiency, where the application automates initial screening, reducing time in literature search and selection; Comprehensive Coverage, providing greater access to studies and overcoming limitations of manual searches; Facilitation of the Review Process, allowing researchers to gather and review a larger set of abstracts for deeper analysis; and Increased Accessibility, facilitating discovery and access to a wide range of publications, including those beyond immediate research parameters. These clearly defined points summarize the benefits of the developed application and demonstrate a structured approach to the conclusion section.
The document suggests, as future work, the addition of new features and resources (such as text mining) to the system. However, it could be interesting to suggest, as future work, case studies or practical examples of how web scraping has improved literature research processes, which could enrich the practical utility of the document. Furthermore, as future work, comparing different web scraping tools or methodologies for bibliographic research applications could provide a broader perspective on the topic.

Cite this review as

·

Basic reporting

1. The article content is clear and easy to understand.
2. The authors have given a detailed background study and the literature references
3. Too lengthy paragraphs. The quality of Figures 2, 5, 6, and 9 has to be improved
4. Authors have provided sample coding screenshots only, no experimental result analysis
5. The result section has to be improved.

Experimental design

1. The authors have focused on developing a web scraping application for searching and extracting the required and relevant research articles from the World Wide Web based on the queries given by the user
2, Authors have highlighted the contributions of their proposed application. No research question is defined and the objectives of the research have not been given.
3. Plenty of web crawling and web scraping applications are available. Authors should compare their proposed application with the existing applications and provide uniqueness in the proposed work
4. On page no 5, lines 258-259 ”Exact match and pattern matching approaches were used". This should be explained in detail and their performances have to be analyzed
5. Authors have provided information about how the users will use the Web crawler chatbot. It is good to share after collecting the huge volume of website information by the web crawler and then how it is filtered to get the optimal search results.
6. Lines 291-292 state that "Before browsing the obtained pages, an algorithm has been developed to check that there is no problem in scraping the article....". The pseudocode of the developed algorithm and its working are required
7. Lines 298-299 given that "to download and scrape the page, unique algorithms have
been developed for each of the article sites where the application will run". What are the unique algorithms developed? Pseudocode, Working of these algorithms and their efficiencies are required.
8. Line 311, "We have developed similar algorithms for scraping on Springer and ACM sites". Here also, the algorithms were not discussed. In the same way, the algorithm for IEEE, an algorithm for ResearchGate also needed
9. The database, structure, number of features, and instance details are also required
10. Authors should provide the NLP/text mining algorithms used for keyword relevance, publication date, citation analysis, and journal ranking

Validity of the findings

1. The results and discussion section focused on elaborating on the working of the web scraping app instead of experimenting with the algorithms, techniques, and methods used for different stages of the work'
2. Various performance factors are to be analyzed for each stage and their performance has to be identified and visualized by using tables, charts, and graphs
3. In the outputs section, a table has to be created to provide the time taken to access the details from different platforms
4. Normally, in the research article screenshots are not required. Hence, Figures 10 to 13 may be removed
5. The experimental result section should discuss the efficiency of the web crawling, web scraping, and relevant search results from Springer, IEEE, and ResearchGate. The accuracy of these algorithms have to be identified

Cite this review as

Reviewer 3 ·

Basic reporting

The authors employ clear and unambiguous, professional English throughout the manuscript, ensuring accessibility to a wide academic audience. The introduction and background sections effectively set the context, highlighting the importance of literature search in academic and research settings. They provide a well-referenced and relevant literature review, demonstrating the current landscape and identifying gaps that the proposed application addresses.

Experimental design

The experimental design presented in the manuscript is generally robust and well-structured, though there are areas for improvement. The primary research question is clearly defined, and the methods chosen to address this question are detailed comprehensively. However, more information is needed regarding the sources of the data and the accuracy of this data. While the explanation of the methodology is sufficiently detailed, technical terms and processes should be made more accessible to readers who may not be familiar with the subject matter. Additionally, the ethical and legal aspects of the research design have not been adequately addressed, and these areas require further emphasis. Overall, the experimental design could be further strengthened with additional information that enhances the study’s validity and reliability.

Validity of the findings

The validity of the findings in the manuscript is generally supported by a robust and well-controlled data analysis. The statistical methods used are appropriate for the research question and are applied correctly, ensuring that the results are reliable. The data is presented clearly, with relevant figures and tables that help to illustrate the key points. However, the manuscript could benefit from a more detailed discussion on potential limitations and biases that may affect the findings. Additionally, it would be helpful to include a comparison with existing literature to contextualize the results and highlight the study’s unique contributions. Overall, while the findings are valid and well-supported, addressing these additional aspects would further strengthen the credibility and impact of the study.

Additional comments

Additionally, I have a few suggestions regarding the overall organization and flow of the manuscript. Firstly, providing a brief overview of the structure of the paper at the end of the introduction would help readers understand the layout and follow the content more easily. In the methods section, a detailed list of the software and hardware tools used, along with the reasons for their selection, would enhance the reproducibility of the study. In the discussion section, it would be beneficial to elaborate on how the findings can be translated into practical applications and what new questions they raise for future research. Finally, clearly stating the limitations of the study and discussing how these limitations might impact the findings would strengthen the manuscript. These additions would significantly enhance the scientific contribution and readability of the paper.

Cite this review as

---

## Round 0.2 · accepted · Accept

Dear authors, we are pleased to verify that you meet the reviewer's valuable feedback to improve your research.

Although in the last round there was only feedback from one of the previous reviewers, after my verification, the manuscript is ready to be accepted.

Thank you for considering PeerJ Computer Science and submitting your work.

Reviewer 3 ·

Basic reporting

The revisions have been effectively addressed, demonstrating clear and professional English throughout the manuscript. The literature references and field background are sufficient, providing the necessary context. The article structure, including figures and tables, is well-organized, and relevant raw data is shared. The results are self-contained, clearly aligned with the hypotheses, and include well-defined terms and detailed explanations.

Experimental design

The experimental design presents original primary research that aligns with the journal's Aims and Scope, addressing a well-defined and relevant research question. The study clearly identifies and aims to fill a significant knowledge gap in the field. A rigorous investigation was conducted, adhering to high technical and ethical standards. The methods are thoroughly described, providing sufficient detail for replication. Overall, the research contributes valuable insights and meets the required scientific rigor.

Validity of the findings

The findings are valid, with all underlying data provided, demonstrating robust, statistically sound, and well-controlled results. The conclusions are clearly stated, directly linked to the original research question, and appropriately limited to the supporting results, encouraging meaningful replication where beneficial to the literature.

Additional comments

The revisions made have been generally deemed sufficient.

Cite this review as